# Development of a Kairomone-Based Attractant as a Monitoring Tool for the Cocoa Pod Borer, *Conopomorpha cramerella* (Snellen) (Lepidoptera: Gracillariidae)

**DOI:** 10.3390/insects13090813

**Published:** 2022-09-06

**Authors:** Jerome Niogret, Paul E. Kendra, Arni Ekayanti, Aijun Zhang, Jean-Philippe Marelli, Nurhayat Tabanca, Nancy Epsky

**Affiliations:** 1Mars Wrigley, Nguma-Bada Campus, James Cook University, Smithfield, QLD 4878, Australia; 2Centre for Tropical Environmental & Sustainability Science, Nguma-Bada Campus, James Cook University, Smithfield, QLD 4878, Australia; 3Subtropical Horticulture Research Station, USDA-ARS, Miami, FL 33158, USA; 4Mars Cocoa Research Centre, Mars Wrigley, Tarengge, Luwu Timur 92971, Sulawesi Selatan, Indonesia; 5Invasive Insect Biocontrol and Behavior Laboratory, United States Department of Agriculture, Agricultural Research Service, Beltsville, MD 20705, USA; 6Plant Sciences Laboratory, Mars Wrigley, Davis, CA 95616, USA

**Keywords:** cocoa pod borer, *Conopomorpha cramerella*, kairomone, pheromone, lure development, monitoring tool

## Abstract

**Simple Summary:**

We have investigated the efficacy of five potential kairomone attractants for the cocoa pod borer (CPB) in Indonesia. Among them, the organic lychee flavor extract in a primitive formulation was the most attractive and was competitive with the commercial pheromone lures. An improved kairomone lure formulation based on specialty membrane and the concentrated extracts captured as many CPB males during a 4-week period and was significantly active for as long as 28 weeks. The long-life pheromone lure may be particularly useful in monitoring large-scale cocoa farms and developing new mitigation technologies that would necessitate high longevity powerful attractants. It should provide a cheaper alternative to the pheromone lures.

**Abstract:**

The cocoa pod borer (CPB), *Conopomorpha cramerella*, is a major economic pest of cocoa, *Theobroma cacao*, in Southeast Asia. CPB monitoring programs currently use a costly synthetic pheromone lure attractive to males. Field trapping experiments demonstrating an effective plant-based alternative are presented in this study. Five lychee-based products were compared for their attractiveness to CPB males. The organic lychee flavor extract (OLFE), the most attractive product, captured significantly more CPB as a 1 mL vial formulation than unbaited traps, while being competitive with the commercial pheromone lures. Additional experiments show that a 20 mL membrane OLFE lure was most effective, attracting significantly more CPB than the pheromone. When the kairomone and pheromone lures were combined, no additive or synergistic effects were observed. Concentrating the OLFE product (OLFEc) using a rotary evaporator increased the lure attractiveness to field longevity for up to 28 weeks; in contrast, pheromone lures were effective for approximately 4 weeks. The 20 mL concentrated OLFE membrane lures should provide a cheaper and more efficient monitoring tool for CPB than the current commercial pheromone lures.

## 1. Introduction

The cocoa pod borer (CPB) *Conopomorpha cramerella* (Snellen) (Lepidoptera: Gracillariidae) is a moth species endemic to Southeast Asia. It is thought that CPB initially utilized native host trees of the Sapindales order, including rambutan *Nephelium lappaceum*, Fiji longan *Pometia pinnata*, and langsat *Lansium parasiticum*, before becoming a pest of economic importance for cocoa, *Theobroma cacao* L. (Malvales: Malvaceae) in Indonesia, the Philippines, Malaysia, and Papua New Guinea. The pest’s devastating impact on cocoa farms is largely responsible for the drastic decline in cocoa production in Malaysia and other areas of Southeast Asia. It is estimated to be directly and indirectly responsible for about USD500 million in annual losses in Indonesia alone [1]. 

Several control measures have been implemented over the past decades to combat CPB infestation, with limited success. Biological control measures were found to have a minimal impact; parasitoids and entomopathogens did not provide enough efficacy or were not economically viable [2,3,4], while predation and disturbance using black ants did not demonstrate promising results [5]. Preventing CPB oviposition on cocoa pods can be achieved successfully by covering the fruits with plastic bags or sleeves, but many farmers consider this method too labor-intensive [6,7]. Complete and synchronic harvesting of mature pods can significantly reduce pest populations, but only temporarily, due to reintroductions from neighboring farms or alternative hosts [8]. At present, the most effective method for the control of CPB is still the use of pesticides, but the cost/efficiency ratio remains poor due to the pest’s lifecycle (i.e., the larval stages are protected within the pod and are not exposed to topical insecticides) [9,10,11]. More sustainable control methods for CPB should be developed due to increasing concerns about pesticide residues in chocolate. Meanwhile, pesticide applications directly related to the phenology of the crop and the CPB population density should be prioritized over regular and agenda-based systematic applications without knowledge of the pest abundance in the field. 

The CPB female pheromone composition has been identified. A blend of four components ((*E*,*Z*,*Z*)- and (*E*,*E*,*Z*)-4,6,10-hexadecatrienyl acetates and the corresponding alcohols in a ratio of 40:60:4:6) was found to capture more males than traps baited with virgin females in field tests conducted in East Malaysia [10]. Additional testing found that the synthetic pheromone was less effective in field tests conducted in West Malaysia, indicating possible strain differences affecting lure efficacy [12]. Zhang et al., [13] found no regional differences in field tests conducted in Malaysia and Indonesia, but cited lack of commercial quantities and problems with quality control of synthetic pheromone components as issues that may limit the lure efficacy. Our experience indicates that lures obtained from various sources were highly variable in efficacy, longevity, and pricing [14]. Therefore, research was initiated to identify kairomones that could be used as an alternative and more reliable monitoring tool for CPB.

Volatile chemicals released by host plants are typically key components used by female insects to locate and assess oviposition sites [15]. Host plant volatiles may also be integrally involved in synergizing sex pheromones [16]. Thus, combining kairomones and pheromones can be critical for reproductive success in phytophagous insects. Recent studies on the grape berry moth, *Paralobesia viteana* (Lepidoptera: Tortricidae), have shown that both males and females were attracted to traps baited with host plant volatiles, and these traps captured significantly more moths than traps baited with the sex pheromone [17]. Integrating pheromonal compounds with female-attractive plant kairomones permitted the combination of mating disruption and female removal technologies in *Cydia pomonella* (Tortricidae) [18,19]. Additionally, there are cases in which only host kairomones are involved in communication, underlying the importance of an insect’s ambient chemical environment. For example, no pheromones are known for the redbay ambrosia beetle, *Xyleborus glabratus* (Coleoptera: Curculionidae), which vectors the fungal pathogen of laurel wilt disease; rather, dispersing females are attracted to volatile sesquiterpenes emitted from host trees [20]. Similarly, with the small hive beetle, *Aethina tumida (Coleoptera: Nitidulidae)*, an invasive pest that is devastating honeybee colonies in the US, the only known attractants are host-produced volatiles [21,22,23]. Similar to pheromones, some examples of sex-biased responses to kairomone attractants exist. Males of the codling moth, *Cydia pomonella* (Lepidoptera: Tortricidae), are more attracted to pear ester than females [24], sometimes exclusively so [25,26]. For CPB females, Niogret et al., [27] demonstrated clear preferences for oviposition on cocoa pods compared with alternative host fruits. These observations strongly support the importance of plant volatiles in the host recognition process of female CPB. However, no information is yet available regarding whether male CPB respond to any host kairomones.

In initial studies investigating volatile chemicals emitted from CPB hosts, we identified a male-specific putative kairomone attractant based on an extract from the lychee fruit, *Litchi sinensis* Sonn. (Sapindales: Sapindaceae). In this report, we summarize a series of field tests that evaluated the efficacy of five sources of lychee extract. The most attractive product was then compared with CPB capture from pheromone-baited traps. Finally, field tests were conducted to study the potential additive and/or synergetic effect with the pheromone and product modification to improve CPB attraction and longevity. Developing a lure based on chemicals derived from fruit extracts should be substantially cheaper than one based on complex synthetic pheromone blends.

## 2. Materials and Methods

### 2.1. Materials Tested

Five lychee-based products (Nature’s Flavors Inc., Orange, CA, USA) were tested in the field: organic lychee flavor extract (OLFE), organic lychee flavor powder (OLFP), organic lychee love fragrance emulsion (OFELL), natural lychee flavor emulsion for high heat (NLFE), and natural lychee flavor concentrate (NLFC). The kairomone lure formulations consisted of an ‘Eppendorf formulation’, made in an Eppendorf tube (Thermo Fisher Scientific, Waltham, MA, USA) with a 1 mm hole on the cap filled with 1 mL kairomone extract, and a ‘specialty black membrane formulation’, composed of a black membrane (90 × 105 mm; 0.01 mm wall) (Thermo Fisher Scientific, Waltham, MA, USA) containing a piece of corrugated cardboard (65 × 90 mm), filled with 20 mL of extract, and sealed with a heating vacuum sealer.

USDA-ARS (Beltsville, MD, USA) provided pheromone lures (0.1mg of pheromone blend of (*E*,*Z*,*Z*)- and (*E*,*E*,*Z*)-4,6,10-hexadecatrienyl acetates and the corresponding alcohols in a ratio of 40:60:4:6) in polyethylene vials (26 × 8 × 1.5 mm thick; Just Plastic, Norwich, United Kingdom) and used as a positive control. The pheromone blend was prepared as described in Zhang et al., [14]. The second source of pheromone lure was provided by Alpha Scents Inc. (West Linn, OR, USA) using the same ratio as Vanhove et al., [28], but with a slight variation in their polyethylene vials (LDPE microcentrifuge vial, Thermo Fisher Scientific, Waltham, MA, USA). Both kairomone and pheromone lures were hung above the sticky liners of the delta traps. 

### 2.2. Experimental Sites

Field experiments were conducted from November 2016 to December 2020 in three locations differing by the presence of shading trees, the cocoa clones/hybrid used, and the management practices. Sites 1 and 2 were two local private cocoa farms, Insitu and Pepuro, located near Tarengge, East Luwu, Indonesia (−2.545376, 120.792307). Plantings at these sites were composed primarily of cocoa clones PBC123 and BR25 that were planted 3 m apart within a row, with 3 m spacing between rows, irregularly shaded by durian and banana trees. The trees were not regularly pruned but were treated with unknown pesticides and without artificial irrigation. Site 3 was located at the Mars Cocoa Research Center (MCRC) in Tarengge, East Luwu (−2.5574.5, 120.798752) and contained trees of the cocoa clone MO1. No pesticide was applied during our field experiments, but the trees were irrigated. 

### 2.3. Experimental Design

Field trials were conducted using white plastic delta traps (20 × 24 × 11 cm) (ISCA Technologies Inc., Riverside, CA, USA) hung on PVC poles 1 m above the tree canopy. A white sticky liner was inserted into each trap to retain the captured insects. The samples were first tested in Eppendorf tubes with a lid pierced with a 1 mm hole before being hung in the delta trap using a wire. Each experiment lasted 4 weeks (unless specified otherwise), traps were checked weekly, and the number and sex of moths captured were recorded. CPB individuals captured were sexed in the field based on observing their external genitalia, as described in [29]. Undetermined specimens were brought to the lab for further observation under an Amscope 3.5X-0.90X Track Stand Stereo Zoom binocular microscope (Amscope, Irvine, CA, USA). Females have a characteristic ovipositor with hairy anal papillae, while males present a darker and wider caudal segment with a hair pencil. 

### 2.4. Field Experiments

Experiment 1 was a 2-week field experiment conducted at site 1 to compare captures with OLFE, NLFC, NLFE, OFELL, OLFP, and an unbaited control trap. One mL of each sample was added to an Eppendorf tube (Thermo Fisher Scientific, Waltham, MA, USA) perforated by a 1 mm diameter hole in the lid. When the sample was in powder form, it was diluted into 1 mL of ethanol 70% before being added into the tube. The test contained four replicates of each of the six treatments, each replicate consisting of a row of traps with treatments randomized and 12 m between traps within a replicate and between replicates.

Experiment 2 compared the number of moths captured with OLFE lure in the Eppendorf formulation with those captured with two sources of CPB pheromone lures (USDA, Beltsville, MD, USA and Alpha Scents, Inc, West Linn, OR, USA) at site 1. The test contained ten replicates of each of the four treatments, each replicate consisting of a row of traps with treatments randomized and 12 m between traps within a replicate and between replicates.

Experiment 3 compared the attractiveness of concentrated OLFE (OLFEc) with its original OLFE version. The product was concentrated by removing the ethanol solvent using a rotary evaporator Heidolph Hei-VAP (Heidolph, Elk Grove Village, IL, USA) at 50 rpm and 60 °C for 60 min. The concentrated OLFE was then tested against the original OLFE, the pheromone lure (USDA), and an unbaited control. Two field tests (Experiment 3, test 1 and 2) consisted of 10 replicates (site 1) and 5 replicates (site 3) for each of the three treatments, each replicate consisting of a row of traps with treatments randomized, and 12 m between traps within a replicate and between replicates. For each test, the kairomone lures consisted of 20 mL of OLFE or OLFEc formulated in a ‘specialty black membrane formulation’ (90 × 105 mm; 0.01 mm wall) (Thermo Fisher Scientific, Waltham, MA, USA), containing a piece of corrugated cardboard (65 × 90 mm), and sealed with a heating vacuum sealer. The ‘specialty black membrane formulation’ was selected based on its efficacy during a preliminary field screening of various lure formulations. Because the population density in Experiment 3 was relatively low and the two field experiments showed the same tendency, we added the two experiments together as a single experiment with fifteen replicates over a 4-week period.

Experiment 4 consisted of two field tests conducted over 4 weeks in sites 2 and 3 to test the potential additive or synergetic effect of OLFE and the pheromone lures. The 20 mL plastic membrane OLFE lures were deployed alone in the delta trap or in addition to the pheromone lures inside the same trap. Those two treatments were compared with the pheromone lure alone and with the unbaited traps. Five replicates (site #) and ten replicates (site 3) for each of the four treatments, each replicate consisting of a row of traps with treatments randomized, and 12 m between traps within a replicate and between replicates.

Experiment 5 was conducted in site 2 in June 2019 to assess the longevity of OLFEc attractiveness (specialty black membrane formulation) to CPB males compared with pheromone lures and unbaited traps. Five replicates for each of the three treatments, each replicate consisting of a row of traps with treatments randomized, and 12 m between traps within a replicate and between replicates. The number of captures was recorded, and the traps were rotated weekly until there was no significant difference in captures between treatments and control. Neither kairomone nor pheromone lures were replaced in this experiment. The experiment ended in December 2019.

### 2.5. Statistics

Kruskal–Wallis ANOVA by Ranks were performed to compare the number of male CPB captures per treatment in the field experiments. Results were presented as mean ± SD, unless otherwise specified. The effect of the lure type on the number of moth captures overtime in the longevity experiment was analyzed using a generalized linear model (glm) with time used as a continuous predictor (Statistica 12^®^, Dell Inc., Tulsa, OK, USA).

## 3. Results

Experiment 1 shows that OLFE captured more males than the other four extracts and the unbaited control (H_5,24_ = 9.160, *p* = 0.062). The differences were insignificant due to the low trapping number during the low CPB population density months. On average, the Eppendorf formulation lure baited with OLFE form captured 1.4 ± 0.5 males/wk during the initial two-week trial compared with 0.4 ± 0.5 males/wk with the controls (Figure 1). A total of 24 insects were captured in this experiment. 

In Experiment 2, both suppliers’ pheromone lures and the OLFE lures were all significantly more attractive to CPB than the unbaited traps (H_3,40_ = 20.707; *p* = 0.001). The USDA pheromone lure captured the most, with an average of 2.3 ± 1.2 males/wk. The pheromone lures from Alpha Scents, Inc. captured an average of 1.5 ± 1.5 males/wk, which was not significantly different from the OLFE lures (1.2 ± 1.0 males/wk) (Figure 2). A total of 199 insects were captured in this experiment.

In Experiment 3, the concentrated extract (OLFEc) was competitive with the USDA pheromone lures. At the Insitu farm (site 1), there was no difference in captures between OLFEc and pheromone lures, but the pheromone caught more moths than OLFE (H_3,40_ = 23.547; *p* = 0.000). All three lures captured more males than the control traps. In the MCRC field (site 3), captures with OLFEc were greater than captures with the pheromone lure, but the treatment differences were insignificant (H_3,20_ = 7.357; *p* = 0.061). Because the population density was relatively low and the two field experiments showed the same tendency, we added the two experiments together as a single experiment with fifteen replicates over a 4-week period. When tests 1 and 2 were pulled together, no significant differences were noticeable between the concentrated kairomone and the pheromone lures, while the kairomone lure attracted more than the unbaited traps (H_3,60_ = 24.748; *p* = 0.000) (Figure 3). A total of 389 insects were captured in this experiment. 

In Experiment 4, the combination of pheromone and kairomone lures within a single trap did not produce any additive or synergetic effects, but the opposite shut-down effect was observed in both field experiments (H_3,20_ = 7.363; *p* = 0.061, and H_3,9_ = 14.002, *p* = 0.003, respectively). Because the population density was relatively low and the two field experiments showed the same tendency, we added the two experiments together as a single experiment with fifteen replicates over a 4-week period.

When presented in tandem, the two lures captured significantly fewer males (1.1 ± 0.7 males per week) than when the pheromone or kairomone lures were presented alone (2.3 ± 1.4 and 2.1 ± 1.4, respectively) and did not attract more males than the unbaited traps (1.1 ± 0.9) (H_3,59_ = 11.327; *p* = 0.010). No difference between the kairomone and the pheromone lures was recorded in this trial. A total of 392 insects were captured in this experiment. 

In Experiment 5, the concentrated OLFE started to capture significantly more males than the pheromone lures and the unbaited traps at week 5 (H_2,15_ = 9.129; *p* = 0.010) until week 28 (almost 7 months) (glm, F(2,48.21), *p*=0.000). Over the 28-week test, OLFEc attracted an average of 143.0 ± 24.4 males per trap compared with 29.6 ± 3.5 males for the pheromone lures and 15.6 ± 5.0 males for the unbaited trap (H_2,15_ = 12.522, *p* = 0.002) (Figure 4). Lures were not replaced in this experiment. A total of 941 insects were captured in this experiment.

## 4. Discussion

Farmers in Sulawesi, Indonesia, and other cocoa-producing regions use prophylactic pesticide applications for CPB control. In Malaysia, for instance, farmers routinely apply pesticides every two weeks. The high frequency of these applications is not based on the pest population density. Developing an efficient low-cost CPB lure would facilitate population monitoring and contribute to more sustainable agricultural practices for cocoa production. The ability to assess population levels would allow cocoa producers to implement control strategies on an as-needed basis, thus reducing costs and minimizing detrimental effects on the environment. An efficient monitoring tool, particularly in combination with a spatial statistical analysis of trap capture, would also help in understanding male flight patterns, phenology, and changes in spatial distribution, improving integrated pest management in conventional and organic orchards. This approach would help direct management decisions, including targeting pesticide applications to foci of infestation, applying pesticides only when threshold levels are reached, and implementing trap crop systems to reduce the impact on cacao [30,31]. The current CPB pheromone lures provide such a monitoring tool. Unfortunately, the only two commercially available sources are expensive for a farmer’s standard and have limited longevity [14]. 

Our results show that OLFE is the most attractive product for detecting CPB males among the extracts tested. In multiple field experiments using the Eppendorf vial formulation, OLFE consistently captured more males than the unbaited control traps, demonstrating the potential of OLFE for the development of a CPB monitoring lure. Kairomone-based lures are often a concern due to the potential attraction of non-target insect species, which usually does not occur with lures containing species-specific sex pheromones [32]. In our study, captures of non-target insects and female CPB were negligible in the OLFE-baited traps. 

OLFE exhibits a similar degree of species specificity and potency as the synthetic pheromone blend for male CPB. Field experiments indicate that only male CPB are attracted to OLFE, a putative host plant-derived product. However, the physiological state of those attracted males was not determined; whether responsive males were sexually immature or mature, virgin, or mated is unknown. Little is known about the mating process in CPB and, in general, about how mating affects the behavioral response to olfactory cues in male insects [33]. In the males of *Agrotis ipsilon* (Lepidoptera, Noctuidae), mating has been shown to trigger behavioral inhibition, characterized by decreased sensitivity to sex pheromones and an increased response to plant odor [34,35]. There are other examples within the Lepidoptera where the attraction of both sexes varies depending on the insect’s physiological state. With *Spodoptera littoralis* (Lepidoptera: Noctuidae), virgin females were more attracted to floral components, whereas mated females were more attracted to cotton leaf volatiles [36]. The authors concluded that virgin females were seeking food sources, while mated females were seeking oviposition sites. Virgin male *S. littoralis* were attracted to both floral and host leaf volatiles initially to locate food resources and mating sites, respectively, but attraction to the volatiles associated with mating sites disappeared after mating [37]. In several species, host volatiles mediate male attraction to mating sites in advance of female emissions of sex pheromones or can synergistically increase the attraction of males when they co-occur with the pheromones [38,39,40,41,42,43]. It is currently unknown whether the reproductive status of CPB males modulates their attraction to OLFE. Further research is needed to understand better the relationship between male CPBs’ physiological state and attraction to OLFE.

Initially, compared with the pheromone lures, OLFE in the Eppendorf vial formulation captured only about half the number of CPB males as the USDA lures (which are not available commercially), but caught an equivalent number of males as the lure developed by Alpha Scents, Inc. When the formulation was improved to the specialty black membrane, OLFE was competitive with the USDA pheromone lures. This capture improvement was independent of the loading dose of OLFE [44]. 

Although multi-component lures have been effective for other agricultural pests (e.g., codling moth [18,19]), when the CPB pheromone and the OLFE lures were combined, no additive or synergetic effects were demonstrated. The number of males captured was lower when the lures were deployed concurrently, compared with the numbers caught by either lure deployed separately. Both lures attracted male CPB exclusively; no females were captured in the field trials. The significant decrease in male captures observed with the lure combination could indicate repellency or a disruption effect. This hypothesis needs further verification, as the synergy experiment was performed during a season with low population densities. We cannot exclude that a repellent or a disruption effect resulted from an overdosage of pheromone precursors potentially present in the OLFE. A preliminary chemical analysis of the OLFE extract did not detect any of the four known pheromone components or precursors, but further study is needed to understand the interaction between the pheromone and the kairomone attractants. 

The attractiveness of OLFE was enhanced by concentrating it using a rotary evaporator to obtain OLFEc. The captures with OLFEc were equal to or better than captures with the USDA pheromone lure over a 4-week period; however, the difference was only statistically significant in one test. However, the key advantage of the kairomone lure (OLFEc) is field longevity. In the 28-week field trial, significantly higher captures were observed with OLFEc at week 5 and the attraction of male CPB continued for nearly 7 months. The attractiveness of the pheromone lures appeared to wane after about week 4, while the attractiveness of the OLFEc appeared to increase after that period. Niogret et al., [14] have shown that the pheromone lures would have to be loaded with 1 mg of pheromone blend to reach comparable longevity.

Identifying the volatile chemical constituents emitted from OLFE would greatly help our understanding of male CPB attraction to this substrate. Isolation, identification, and quantification of chemical emissions from OLFE will be the primary goal of future research. Fractional distillation of the attractive substrate, followed by binary choice bioassays and electroantennography with individual fractions, may lead to identifying key attractive components. This information will clarify the roles of specific volatiles in attracting male CPB, thereby directing future work in optimizing the development of a kairomone lure for detecting and monitoring pest CPB populations. 

## Figures and Tables

**Figure 1 insects-13-00813-f001:**
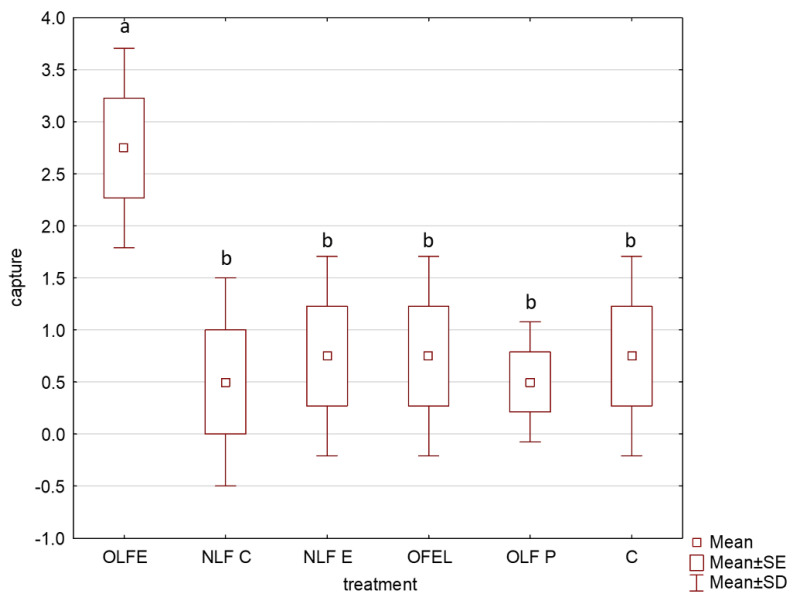
Number of male CPB captured with various lychee-derived products in a two-week field test. OLFE—Organic Lychee Flavor Extract, NLFC—Natural Lychee Flavor Concentrate, NLFE—Natural Flavor Lychee Emulsion, OFELL—Organic Fragrance Emulsion Lychee Love, OLFP—Organic Lychee Flavor Powder. Four replicates for each treatment were tested. Boxes presenting different letters are significantly different.

**Figure 2 insects-13-00813-f002:**
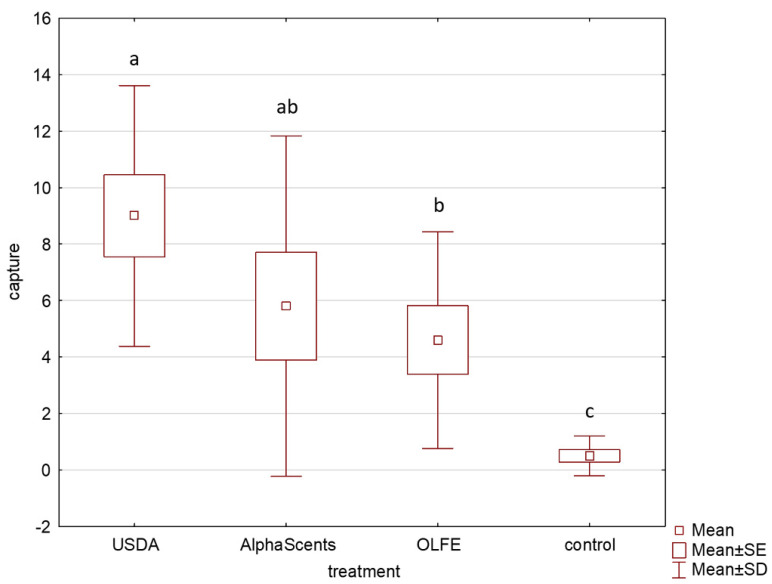
Number of male CPB captured with OLFE (Eppendorf formulation), two sources of pheromone lure (USDA and Alpha Scents, Inc.), and unbaited control traps in a four-week field test. Ten replicates per treatment were used in this experiment. Boxes presenting different letters are significantly different.

**Figure 3 insects-13-00813-f003:**
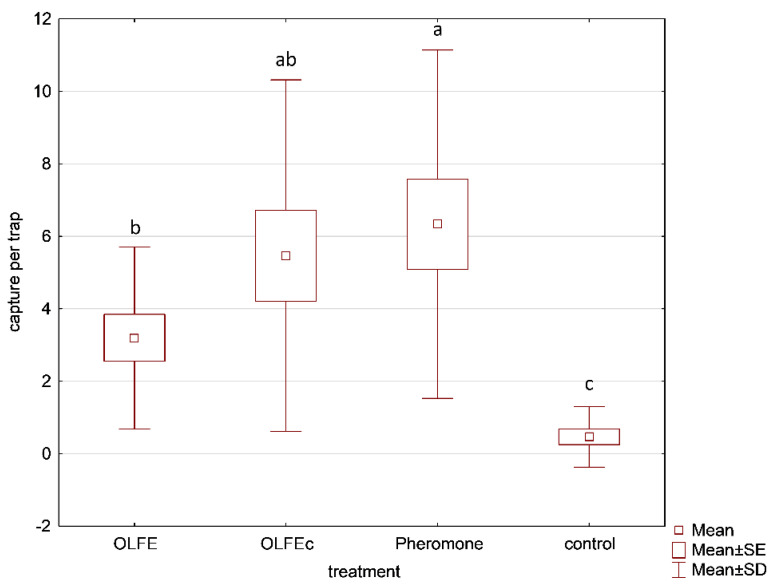
Number of male CPB captured in delta traps baited with OLFE, concentrated OLFE (OLFEc), pheromone lures (USDA), and unbaited control traps. Tests were conducted at the Insitu field site and the MCRC site with a total of fifteen replicates per treatment. Kairomone treatments consisted of 20 mL samples in the specialty black membrane formulation. Boxes presenting different letters are significantly different.

**Figure 4 insects-13-00813-f004:**
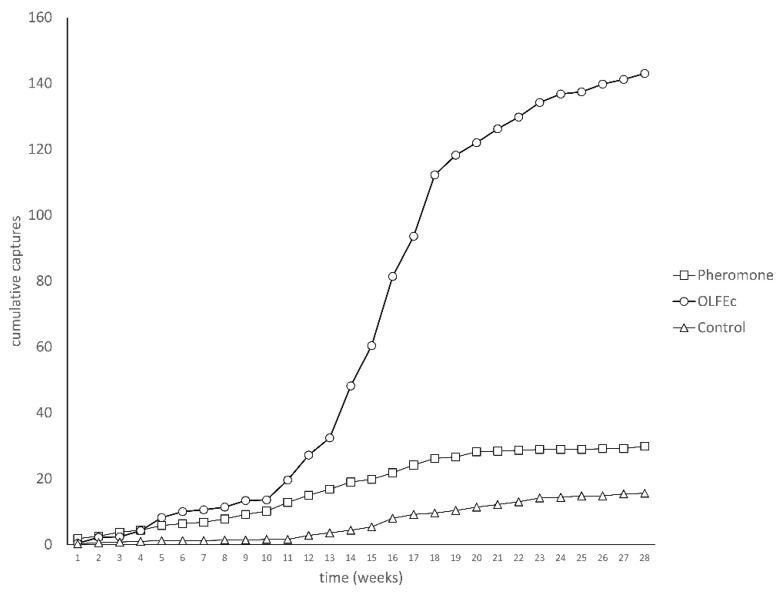
Capture of male CPB using a single lure of pheromone and OLFEc compared with unbaited traps over a period of 28 weeks. Five replicates per treatment were used in this experiment. The experiment continued until no significant differences were recorded between treatment and control.

## Data Availability

The data supporting this study’s findings are available from the corresponding author, Jerome Niogret, upon reasonable request.

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
