# Peer review of "Development of a Kairomone-Based Attractant as a Monitoring Tool for the Cocoa Pod Borer, Conopomorpha cramerella (Snellen) (Lepidoptera: Gracillariidae)"

_insects, 2022, doi:10.3390/insects13090813_

Round 1

Reviewer 1 Report

The paper submitted by Niogret et al. reported that a new kairomone-based attractant is very useful for monitoring the population of the cocoa pod borer. The authors compared the attractiveness of five lychee-based lures to the cocoa pod borer males with the commercial pheromone lures, which had been demonstrated attractive to CPB males. The authors found the product of concentrated organic lychee flavor extract, OLFEc, were most attractive, and showed up to 28 weeks longevity in attractiveness in the field, which is much longer than four weeks of pheromone lures. Therefore, the authors developed a much effective and much cheaper attractant as a tool to trap and monitor CPB males. No more suggestion to the authors.

Author Response

Dear Reviewer 1, thank you for your appreciation of the research and the manuscript. We have done a spell check on the manuscript based on the reviewer's recommendation

Best regards,

The authors

Reviewer 2 Report

In this paper, authors present results of a very well executed trapping study that was conducted to evaluate attractiveness of five kairomones to cocoa pod borer (CPB) and to compare it to unbaited control traps and positive control traps baited with commercial CPB sex pheromone lure. The results clearly indicate that one of the kairomones tested, organic lychee flavor extract, was significantly more attractive than the rest of the tested attractants. This kairomone would make a cheaper yet at least equally attractive lure with higher longevity, compared to a currently used pheromone lure.

This paper presents a very important finding that will allow to increase CPB trapping efficiency, which in turn will improve knowledge of biology and population ecology of this pest and help develop better management tactics. The manuscript is very well written and ready for publication, in my opinion, I have only a few very minor comments.

In the Simple Summary I would add CPB in parentheses on line 16, to make line 19 clear. I would also use the organic lychee flavor extract instead of the abbreviation on line 16.

In the Abstract, on line 27 organic lychee flavor extract should be spelled out. If the number of words is an issue, I would remove the Latin names.

The introduction is very detailed, it clearly demonstrates the problems faced by cocoa farmers and the need for better trapping system to improve management, it also provides good background information on kairomone use as attractants in other pest systems.

The Methods section is also well written and detailed. I was wondering about the 12m distance between the traps, was it chosen based on any previous research or based on the plot size?

The Results section is clear and easy to follow; all results are well addressed in the Discussion section and future research needs are outlined. Cocoa is in high demand, but the trees are hard to grow, I am excited to see that the research is being conducted to help the cocoa farmers!

Overall, this is a very important study deserving of publication.

Author Response

Dear Reviewer 2, thank you for your appreciation of the research and the manuscript. We have followed all the reviewer’s recommendations and added

  • (CPB) in line 16
  • Replaced OLFE by “organic lychee flavor extract” in line 19
  • added “organic lychee flavor extract” in line 27
  • The 12m distance is the commonly used inter-trap distance used to study CPB pheromone lure efficacy in [14: Zhang et al 2008] and [15: Niogret et al. 2022]. Because the kairomone lures seem to only attract males CPB, we have decided to test the kairomone efficacy at the same trap density than the pheromone trap density

Best regards,

The authors

Reviewer 3 Report

I have no major concerns with the manuscript by Niogret et al., but a few minor points need to be addressed:

- line 165: At which site was experiment 2 conducted?

- line 178: How did you arrive at the choice of the membrane dispenser?

- line 200, Statistics: Were raw data transformed for ANOVA; if yes, how? Otherwise, a non-parametric statistical test is needed.

- Results: Indicate total catch for each experiment.

- As you mention OLFE is more cost-effective, how does the price of OLFE compare to that of the pheromone?
